# Proteins in Food Systems—Bionanomaterials, Conventional and Unconventional Sources, Functional Properties, and Development Opportunities

**DOI:** 10.3390/polym13152506

**Published:** 2021-07-29

**Authors:** Jan Małecki, Siemowit Muszyński, Bartosz G. Sołowiej

**Affiliations:** 1Department of Dairy Technology and Functional Foods, Faculty of Food Sciences and Biotechnology, University of Life Sciences in Lublin, Skromna 8, 20-704 Lublin, Poland; j.malecki@eurohansa.com.pl; 2EUROHANSA Sp. z o.o., Letnia 10-14, 87-100 Toruń, Plant in Puławy, Wiślana 8, 24-100 Puławy, Poland; 3Department of Biophysics, Faculty of Environmental Biology, University of Life Sciences in Lublin, Akademicka 13, 20-950 Lublin, Poland; siemowit.muszynski@up.lublin.pl

**Keywords:** health, plant protein, animal protein, food

## Abstract

Recently, food companies from various European countries have observed increased interest in high-protein food and other products with specific functional properties. This review article intends to present proteins as an increasingly popular ingredient in various food products that frequently draw contemporary consumers’ attention. The study describes the role of conventional, unconventional, and alternative sources of protein in the human body. Furthermore, the study explores proteins’ nutritional value and functional properties, their use in the food industry, and the application of proteins in bionanomaterials. Due to the expected increase in demand for high-protein products, the paper also examines the health benefits and risks of consuming these products, current market trends, and consumer preferences.

## 1. Introduction

It is estimated that the ever-increasing population growth will reach around nine billion people by 2050 (Figure 1), resulting in huge demand for protein-rich food worldwide. This estimation indicates the potential insufficiency of conventional protein sources in the future, resulting in increased interest in unconventional proteins [1]. Proteins are the basic macronutrient of the human diet. In terms of chemical structure, proteins consist of carbon, oxygen, nitrogen, hydrogen, sulfur, and phosphorus. The properties and functions of proteins depend on their structure [2]. We can distinguish between simple proteins, consisting mainly of amino acids, and complex proteins with other components attached to the amino acids. Proteins are large biomolecules and macromolecules comprising one or more long chains of amino acid residues. A linear chain of amino acid residues is called a polypeptide. A protein contains at least one long polypeptide. The individual amino acid residues are bonded together by peptide bonds and adjacent amino acid residues [3,4]. Protein is considered a key ingredient in the human diet for assessing the body’s needs due to the complex metabolic changes required to run two processes constantly, such as the synthesis and breakdown of the body’s proteins [5]. This essential process is the function of the multithreaded protein metabolism and is commonly known as protein turnover. Proteins are cell molecules that power virtually every function and development program in biology. Surprisingly, many of these critical molecules easily aggregate and accumulate inside living cells through interactions between developed and complex domains. Aggregation may occur incorrectly and lead to disease, but there is growing evidence that the aggregation phenomenon can be regulated by the cell and used to perform important and beneficial biological functions, from molecular scaffolding to memory [6,7].

In industry, proteins are widely used, depending on their specific functional properties. The main functional differences observed in proteins are their varying structural properties [8]. Polypeptide chain modifications and changes in environmental conditions affect the conformation of protein molecules and, thus, their solubility and ability to form or stabilize emulsions and foams [9]. This review article presents the role of proteins in the human body, characteristic of conventional, unconventional, or alternative sources of proteins. In addition, the nutritional value of proteins, their functional properties, and their use in the food industry are examined.

**Figure 1 polymers-13-02506-f001:**
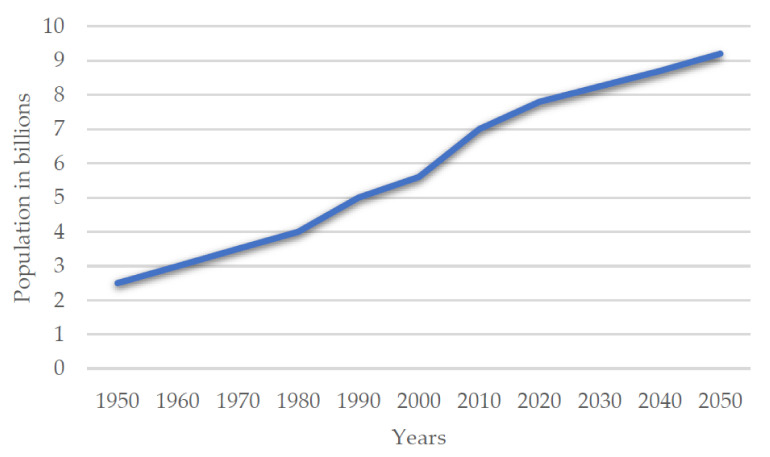
World population growth 1950–2019 with projections to 2050 based on Jensen et al., 2020 [10].

Based on FAO Food Balance Sheet data, it is clear that global meat consumption has increased significantly in recent decades due to the growing population. Henchion et al. 2014 [11] found that overall meat consumption increased by almost 60% between 1990 and 2009. This trend is expected to continue, driven by income growth in countries such as Asia, South America, and the Middle East. While the United Nations and governments are implementing campaigns to reduce the amount of meat consumed [12], global meat consumption is expected to increase by 76% by 2050 compared to 2017 [13]. The global plant-based protein market is projected to grow from USD 10.3 billion in 2020 to USD 14.5 billion by 2025, recording a compound annual growth rate (CAGR) of 7.1% during the forecast period. The major factors driving the growth of the plant-based protein market include growing demand in the food industry, increasing demand for pea-based protein, and the opportunity to expand in potential high-growth markets [14,15]. Consumers make food choices based on traditional food values such as taste and price. However, other food values, such as health, environmental impact, and ethical concerns, now influence consumer decisions. Due to these influences, increasing interest in unconventional proteins (including plant and insect proteins) is expected in the coming years [16].

Proteins are subject to constant interactions related to the influence of other nutrients and energy metabolism. They are the basic structural and functional components of every human body cell, responsible for gene expression control, and essential for the young organism’s proper growth and development. They are part of many enzymatic systems (as biocatalysts) [14]. Proteins perform the function of transporting oxygen (hemoglobin), iron (transferrin), and retinol (eight-stranded β-barrel proteins that bind extracellular retinoids, such as retinol-binding protein 4 and epididymal retinoic acid-binding protein). Furthermore, actin and myosin are muscle-contractile elements participating in tissue repair and regeneration. Proteins (as antibodies) participate in cellular and humoral immunity processes [17,18]. Proteins function as substrates in synthesizing many hormones and biologically active compounds, such as adrenaline, noradrenaline, thyroid hormones (thyroxine, triiodothyronine), histamine, and serotonin. In addition, they participate in creating biologically active compounds such as purine and pyrimidine bases (components of nucleotides and nucleic acids), choline (phospholipids component), glutathione, creatine, and many other components involved in physiological processes [19]. The unique characteristics of their pure enzyme forms make proteins highly applicable to several chemical transformation reactions in the pharmaceutical industry, such as group protection and deprotection, selective acylation and deacylation, selective hydrolysis, deracemization, kinetic resolution of racemic mixtures, esterification, and transesterification [20].

## 2. Functional Properties of Proteins

Proteins can have surface properties such as the ability to form or stabilize emulsions (interfacial oil/water interface), the ability to create or stabilize foams (interfacial air/water interface), or solubility (combining the connections between water and proteins). In addition, proteins have hydrodynamic properties based on intermolecular interactions, including gelation, texture, and molding sensory properties (taste and smell) [21]. The functional properties of proteins (Figure 2) depend directly on the specific properties of their molecules, such as size, shape, susceptibility to denaturation, flexibility, amino acid composition, hydrophilicity and hydrophobicity, the charge distribution in the molecule, the nature and number of microdomain structures, the ability of the entire molecule or its constituent domains to adapt to changing environmental conditions, and the nature of the interrelationships between different proteins and other food components [22]. The functional properties of proteins are affected by important environmental factors in the protein’s location, such as pH, temperature, pressure, and ionic strength [8]. Proteins form complex systems with other food ingredients that affect the formation of their functional properties, and additionally, technological processes play a significant role in shaping proteins’ functional properties [23]. In most proteins, the majority of hydrophilic functional groups are located on the surface of the molecules. However, the hydrophobic groups are not entirely located inside them. In globular proteins, 40–50% of the molecule’s surface may be occupied by hydrophobic amino acid residues [24]. Their specific distribution in the polypeptide chains affects the surface formation of protein molecules, the ability to create oligomers and micellar structures, and functional properties [25].

Sulfuric amino acids play a significant role in shaping the structure and function of proteins [26], and thiol groups can be oxidized to form intra- and intermolecular disulfide bridges. These interactions change the structure and function performed by the proteins. Cross-bending disulfide bridges stabilize proteins’ tertiary structure and affect their functional properties [27].

### 2.1. Functional Properties of Proteins

Among proteins’ functional properties, solubility deserves special attention. Solubility is considered the basic functional property of food proteins. This property largely determines the applicability of protein preparations in food technology. High solubility is often associated with the protein’s good functional properties, enabling producers to create food products with desired, repeatable, and predictable characteristics [28]. Loss of solubility due to food processing under harsh conditions is often an indicator of denaturation and subsequent protein cross-linking [29]. The solubility of proteins depends on the structure and properties of the solvent, temperature, pH of the environment, concentration and charge of ions, and the nature of interactions with other molecules [30]. Surface hydrophobicity and the resultant electric charge are the most important characteristics of the protein molecule, determining its behavior towards the solvent. Surface hydrophobicity is an indicator of the character of the diverse electrostatic potential of various protein surface fragments, decisive for the protein’s spatial shape and behavior towards polar and apolar solvents [31]. One method for modifying hydrophobic/hydrophilic surface protein molecules is chemical modification such as glycosylation, and enzymatic modification (density cargo), such as dephosphorylation [32]. Protein use helps to maintain high solubility in an environment. This property is particularly significant in an acidic environment. For example, using proteins as additives to juices and beverages prevents coagulation [33].

### 2.2. Surface Properties of Proteins

The phenomenon of proteins stabilizing emulsions or foam is caused by their ability to adsorb at the interface, reduce surface tension, and create a coherent layer around oil droplets or air bubbles [34]. If the surface of the particles is entirely hydrophilic, phase interface adsorption may not occur. However, if it contains only a few hydrophobic residues and interacts with the phases’ surface, adsorption may occur. In other words, adsorption at the interface between air/water and oil/water depends on the statistical probability of collisions of hydrophobic groups located on the surfaces of protein molecules with a phase boundary. Conformational stability, the ability to rearrange at the interface, and symmetry/asymmetry in distributing polar and apolar functional groups influence the membrane adsorption and formation amphiphilicity of protein structures [35]. Due to these properties and their greater ability to provide satiety than carbohydrates and fats, proteins can act as a regulator of gastrointestinal hormones to increase the feeling of saturation and reduce the calories absorbed from meals [36]. The addition of foams with bubbles to food products such as chocolate, cheese puffs, and gelatin foams introduces innovation and reduces the products’ calorific value by increasing their bulk compared to their standard counterparts [37].

Furthermore, aerated gels have applications in the formation of capsular products, the release of taste, the selective supply of bioactive particles, satiety control (similar to foams), and the creation of gastronomic structures [38]. Therefore, proteins can shape the desired texture of a food product, improve water absorption, and prevent syneresis. The gel matrix is retained; immobilized water molecules and other food ingredients such as carbohydrates, polyhydric alcohols, and fibers help to create a stable gel structure for food products [39].

### 2.3. Foaming Capacity

Foams are created by dispersing air bubbles in the liquid phase. Adding protein increases the aqueous phase’s viscosity, increasing the interfacial film’s durability and producing foam [40]. Proteins reduce the surface tension by interacting with the water molecules and air, allowing foam bubbles to form. After adsorption on the surface of the air bubbles, polar amino acid residues on the surfaces of the protein molecules react with the liquid, and non-polar amino acids react with the air, resulting in the formation of a coherent, flexible film around the air bubble’s interphase. The bubbles remain separate from each other because unchanged fragments of the protein molecules connect with them. The 0.1–1 mm diameter air bubbles can comprise up to 99% of the foam’s total volume [41]. The factors influencing foam formation include surface hydrophobicity, the location of hydrophobic amino acid residues on the protein’s surface, thiol groups, cations and anions, carbohydrates, and lipids. The stability of the formed foam depends on the protein’s ability to protect the foam from the effects of gravity and mechanical interactions [42]. Stable foam is usually created at a pH close to the protein’s isoelectric point, when electrostatic interaction forces are the smallest. Processes that increase hydrophobicity improve foaming properties. The protein’s foaming properties can be increased with short periods of heating. For example, thermal denaturation for 30 min at a temperature range of 40–60 °C improves the foaming properties of whey proteins. Optimal heating conditions depend on the type and concentration of protein [43].

### 2.4. Creating Emulsions

Emulsions are dispersion systems consisting of two or more immiscible liquids in the form of a continuous and a dispersed phase (small droplets). When mixing oil and solutions with aqueous proteins, it is preferable to limit contact between them and phase separation. Initially, minimal contact is achieved due to spherical droplets’ formation with energy input from the outside. A stabilizing agent is introduced to facilitate the emulsion’s formation and improve its stability. Proteins, as an emulsifying agent, can be used as a stabilizer [43,44]. Protein-stabilized emulsions ensure minimal contact between hydrophobic groups and water and are energy-efficient as they do not require high energy expenditure during emulsion formation [44]. The time required to form a coherent layer around the oil droplets and establish thermodynamic equilibrium depends on the protein type. With loose, elastic-structured proteins, these phenomena proceed quickly, at medium speed using globular proteins, and slowly with proteins with a compact structure. The droplet size of the dispersed phase characterizes the basic size of the emulsion. The diameter of these drops in food product emulsions varies between 0.2 and 10 mm. The size depends on the emulsion production method, the difference in viscosity between the two phases, the emulsifier used, and the energy input during emulsion formation [45]. Low-quality products contain drops of approximately 10 mm diameter and above. In high-quality products such as mayonnaise, the drops are 2–4 mm [46]. Industry commonly uses the proteins lysozyme, β-lactoglobulin, β-casein, α-lactalbumin, and ovalbumin for emulsifying [47].

In contrast to low-molecular-weight emulsifiers, the structure of proteins may be affected by adsorption. Therefore, a good emulsifier should not only create but also stabilize the newly formed interface. Protein-stabilized emulsions are more stable at a pH other than the isoelectric point values of the proteins. The emulsion’s stability depends on the continuous phase’s viscosity, gravity forces, the resultant charge, and the protein’s structure. In the emulsion preparation device, the energy provided during emulsification mainly determines the extent and nature of changes to the emulsion over time [48]. Furthermore, environmental factors, such as protein concentration, active acidity, oil/water phase ratio, and ionic strength, determine the emulsion’s stability [49]. Changing the protein’s structure can lead to conformational changes and affect its ability to create and stabilize emulsions. The increase in hydrophilicity can play a positive role in shaping the emulsifying properties [50].

### 2.5. Protein-Based Bionanomaterials

In nature, we find many examples of solid and functional bionanomaterials understood as a combination of bio-macromolecules (mainly proteins) with small organic molecules or materials, providing the basis for the production of advanced and highly efficient hybrids (photosystems, metalloenzymes, antenna systems, and bionanocomposites) [51]. Nanomaterials are significant in the developing field of science and economy. Size reduction could result in several new physicochemical properties and many potential applications. Nanotechnology is an innovative technology that uses methods and techniques to obtain materials, elements, and devices with at least one controlled dimension in the nanoscale range of 1–100 nm [52]. Two techniques are used in the production of nanomaterials: top-down and bottom-up. The top-down method consists of reducing the particle size. The bottom-up method considers the construction of new structures based on existing nanoparticles [53]. Using this method, the nanostructures’ building blocks can be atoms, molecules, or nanoparticles, depending on the properties of the final product.

It is possible to obtain a material with the desired properties by changing the size of the building material, controlling the features of its surface and interior, and imposing specific conditions for joining particles into a nanomaterial [54]. Nanotechnology applications apply to all areas of food science (Figure 3), including agriculture, food processing, packaging, safety, nutrition, and nutraceuticals [55]. New approaches are being applied to the development and design of new protein-based bionanomaterials. The significance of using proteins in the production of bionanomaterials goes beyond their intrinsic functionality, as proteins can also be used as highly tunable platforms as a basis for accommodating and binding synthetic materials, suggesting new functions for the hybrid system. In addition to functional and versatile structural proteins as building blocks for design, it should be noted that, compared to other platforms, protein-based materials are ecological, durable, biodegradable, and biocompatible [51].

## 3. The Most Common Sources of Plant and Animal Proteins

Food proteins are essential nutrients required for maintaining various bodily functions and human health. Proteins, especially some traditional plant- and animal-derived protein sources (Figure 4), are essential food ingredients. They are listed and described in this section.

### 3.1. Soy

Soy is the primary plant for protein production, with a global soybean harvest of around 300 million metric tons per year [56]. Soybean has been a food source for many years. It is the globally adopted GM crop, covering 80% of GM crop-growing areas worldwide, corresponding to approximately 100 million metric tons per year. The main threat associated with genetically modified organisms is biodiversity disturbance, resulting from the uncontrolled modification of transgenic organisms released into the environment [57]. Modified varieties can displace traditional plant varieties and reduce the number of certain species. The emergence of agricultural monocultures contributes to the resistance of plant species and insects to the chemical agents controlling them and a significant increase in their population. Crossing transgenic plants with rapid-growth wild plants can lead to the formation of “superweeds” [58]. Therefore, the safety of genetically modified crops and transgenic food products requires detailed analysis and clarification of issues such as the toxicity and health safety of GM plants, the impact of GM plants on other organisms, the allergenicity of food products made from GM plants, biological safety, and resistance to antibiotics [59].

The United States is the largest producer of soy and soy products [60]. Soybeans contain a naturally high protein content (35–40%); soy is widely used in oil production, and soy flour is generated during this process. A significant amount of defatted soy flour serves as animal feed. The remaining flour is used for various kinds of high-protein products intended for human consumption [61]. The relatively high protein content and preferably balanced amino acid composition make soy protein suitable as a substitute for meat and milk proteins in humans’ daily diet [62]. The most common soy products are soy flour, soy protein concentrate, soy protein isolate, textured, and hydrolyzed soy protein. Soy protein concentrate (SPC), containing approximately 65% protein, is obtained from defatted soy flakes free of soluble parts of the cell walls. Soy protein isolate (SPI), with approximately 90% protein, is the most highly refined soy product. It is formed by alkalic extraction and isoelectric precipitation. The textured soy protein produced by the extrusion method resembles the texture of meat [56]. This product’s primary function is an alternative protein source for the complete or partial replacement of animal protein in various food products, particularly in the daily diet of vegans and vegetarians. Isolates, soy protein, concentrate, soy flour, and the textured products derived from them are commonly used in food industry preparations due to their functional properties, such as water and fat binding, emulsifying, foaming, and gelling [63].

### 3.2. Wheat

Wheat is the most cultivated and the most significant crop in the world. It is consumed by over a billion people around the world in various forms [64]. In 2017, the total global production of wheat was 772 million metric tons, with approximately 150 million metric tons used as animal feed. Based on 2017 data, the European Union is the largest wheat producer, with 150.2 million metric tons. China, India, and Russia account for around 41% of the world’s total wheat production [65,66]. Wheat is a more frequent source of protein and calories than any other food due to the global consumption of wheat products [64]. The protein content in wheat varies from 8% to 15%, depending on the variety [67]. The amino acid composition of wheat is quite unbalanced, lacks essential amino acids such as lysine, threonine, and methionine, and processing wheat into various products further depletes essential amino acids [68].

The main reservoir of protein in wheat grains is gluten protein. Gluten has unique functional properties not found in other plant proteins. Gluten creates a coherent, slightly elastic, cross-linked protein structure that allows wheat to be used to produce products such as sourdough [67]. Commonly used gluten, with a protein content of around 80%, is obtained by simply rinsing the flour with water. This method was discovered in the second half of the 19th century. In addition to the conventional method of obtaining gluten, several other methods of obtaining and modifying this protein are used [69]—for example, the chemical and enzymatic modification used in eluted gluten to strengthen its structure. Thanks to available technologies, it is possible to achieve 90% protein content in wheat protein isolates. The main application of gluten is bakery products, pasta, and breakfast cereals. As gluten possesses an efficient ability to abstract fat and water, it is used in the meat and fish industry [70]. Gluten is commonly used as a substitute for meat proteins in foods for vegetarians and vegans. Textured wheat protein, obtained by extrusion, is increasingly used to imitate meat products’ appearance and structure [71].

Gluten-related diseases such as celiac disease and gluten ataxia are rare conditions, affecting less than 1% of the population. Despite the rarity of these diseases, there has been a significant increase in the adoption of a gluten-free lifestyle and the consumption of gluten-free foods over the last three decades [72]. Individuals might restrict gluten from their diets for various reasons, such as improvement of gastrointestinal and non-gastrointestinal symptoms, and because of the perception that gluten is potentially harmful and, thus, restriction represents a healthy lifestyle. Emerging evidence shows that gluten avoidance may benefit some patients with gastrointestinal symptoms, such as those commonly encountered with irritable bowel syndrome [73].

### 3.3. Milk Proteins (Casein)

Protein is an important component of milk as it largely determines its nutritional value and suitability for processing. Cow’s milk contains approximately 3.4% protein on average and is the sum of two main fractions, casein and whey proteins, constituting approximately 80% and 20% protein and nitrogen compounds, respectively [74], and differing in their physicochemical properties. The knowledge of these compounds and their practical use is the basis for producing various milk–protein preparations. The raw material casein is used to produce pasteurized skimmed milk and skimmed milk powder [75]. With either acid or rennet coagulation methods used, casein is distinguished. Acid casein contains, on average, around 88% protein, 1.5% fat, 0.3% lactose, and 2.1% ash. The composition of rennet casein is approximately 82% protein, 1.4% fat, 0.5% lactose, and up to 8.5% ash [76]. Cow’s milk casein contains all essential amino acids in greater amounts than the FAO/WHO standard. When comparing casein with whole hen egg protein, lower contents of isoleucine, lysine, threonine, tryptophan, valine, and sulfur amino acids are apparent [77]. A significant quality-distinguishing feature of caseinate preparations is their functional properties, characterizing how proteins interact with food ingredients to determine their potential practical application. In food processing, the important functional properties of caseinates are solubility, water absorption, viscosity, gelling, fat binding, emulsification, and foaming [78]. Due to their properties, caseinates are used in many food processing industries, such as meat processing, delicatessen production, cereal product production, baking, confectionery, dairy, beverage production, food concentrates, and the preparation of products for special nutritional purposes [79]. In a section of the population, milk proteins can trigger an allergic reaction. Cow’s milk allergy (CMA) is an immunologically mediated reaction to cow’s milk proteins, involving the gastrointestinal tract, skin, respiratory tract, or sometimes multiple systems (systemic anaphylaxis). Its prevalence in the general population is probably 1–3%, highest in infants and lowest in adults [80,81].

### 3.4. Whey Proteins

Due to their favorable physicochemical and biological properties, whey proteins are now perceived as nutrients in the production of dietetic food, physiologically active in the production of functional foods, and structure-forming in traditional and new generation food [82]. Intake of 14 g of whey proteins covers the daily requirement for essential amino acids of a person weighing 70 kg, equivalent to 23 g of casein or 17 g of egg white. Whey proteins are popular due to their biological activity and the possibility of using them to produce so-called functional food [83]. The composition of whey’s dry matter justifies its use as a raw material for further processing. Ultrafiltration densification is increasingly used to reduce the costs of concentrated whey operations. In addition to their nutritional properties, whey proteins offer a wide range of functional properties [84]. The meat industry uses whey proteins to improve taste, texturing, emulsifying, gelling, binding water, and improving nutritional properties. Whey proteins enable partial replacement of meat proteins, or replace soy products and other non-meat additives such as modified starches [85]. Using an appropriate proportion of functional whey proteins (about 35%) and milk powder in yogurt production improves its rheological and taste properties and allows the resignation or restriction of non-dairy thickeners such as gelatin, modified starches, and pectin [86]. In confectionery, whey proteins can be substituted for skimmed milk powder. In addition, whey proteins are used for non-fat mayonnaise production, sauces, and soups. Their advantages are solubility in a broad spectrum of pH and the ability to form gels, bind water, and imitate the taste properties of fats [87].

### 3.5. Egg White Proteins

Eggs are of particular interest from a nutritional point of view, gathering essential lipids, proteins, vitamins, minerals, and trace elements while offering a moderate calorie source (approximately 140 kcal/100 g), great culinary potential, and low economic cost. Indeed, eggs have been identified as the lowest-cost animal source for proteins, vitamin A, iron, vitamin B_12_, riboflavin, and choline, and the second-lowest-cost source for zinc and calcium [88,89]. Egg proteins are distributed equally between the egg white and egg yolk, while lipids, vitamins, and minerals are essentially concentrated in the egg yolk. The relative content of egg minerals, vitamins, and specific fatty acids varies between national references but remains globally comparable when considering major constituents such as water, proteins, lipids, and carbohydrates [90]. The major egg nutrients are very stable and depend on the ratio of egg white to yolk in contrast to minor components affected by several factors, including hen nutrition. As a whole, raw, freshly laid eggs’ water, protein, fat, carbohydrates, and ash represent approximately 76.1%, 12.6%, 9.5%, 0.7%, and 1.1%, respectively [91]. Egg white contains around 10% protein, with many functionally important proteins, including ovalbumin (54%), ovotransferrin (12%), ovomucoid (11%), ovomucin (3.5%), and lysozyme (3.5%), among the major proteins that have high potential for industrial applications, if separated [90]. Simplicity, high method reproducibility, non-toxic chemicals used for separation, and the sequential separation of many proteins are significant criteria for the commercial production and application of egg proteins. The separated proteins are used in the food and pharmaceutical industry, and they can be modified with enzymes to meet the needs of a given industry. Ovotransferrin is used as a metal transporter, antibacterial, or anti-cancer agent, while lysozyme is mainly used as a preservative in food applications [92,93]. Ovalbumin is widely used as a dietary supplement and ovomucin as a cancer inhibitory agent [90]. Ovomucoid is the major egg allergen, but it can inhibit the growth of tumors and therefore is used as an anti-cancer agent. Hydrolyzed peptides from these proteins show good angiotensin I converting enzyme inhibitory, anti-tumor, metal-binding, and antioxidant activity [94]. Therefore, egg proteins and the production of bioactive peptides from these proteins are new areas with many possible applications [95,96]. Furthermore, due to their functional properties, egg proteins are widely used in various branches of the food industry for gelling, coagulating, foam formation, stabilization, and water binding. In addition, they are used in cheese production (ripened cheeses), the meat industry (raw meat and fish, pates, baking, sausages, and canned meat), in confectionery (foam products and meringues), in the production of beer, wine, and mead, and in the fat industry [97].

### 3.6. Gelatin

Gelatin is an animal protein of high purity, produced from collagen and constituting approximately 30% of a protein substance found in high-collagen areas of the animal and human body, especially in bone, cartilage, connective tissue, and skin. The production process transforms collagen into water-soluble gelatin [98]. Collagen determines the physicochemical properties of gelatin, and in particular, the gel strength of the obtained proteins [99]. Collagen is a protein with an unusual amino acid composition. It contains significant amounts of glycine and proline and two amino acids not derived directly from translation in ribosomes: hydroxylysine and significant amounts of hydroxyproline [100]. Regardless of the gelatin production method, the ability to reversibly form gel is its most significant feature. Different species of gelatin have lower or higher gelling capacities. Gelatin serves as a gelling agent, stabilizer, protective colloid, emulsifier, foaming agent, carrier, and binder and is an important auxiliary material in pharmacy. It can be used to produce capsules, liquid medicines, dragées, and granulates [101]. In the food industry, gelatin is a good solution for the production of low-calorie semi-finished products. As a stabilizing and binding additive, it can be used in yogurts, jelly, meat, and fish products. Using gelatin improves the structure in the process of freezing and thawing confectionery products [102].

### 3.7. Protein Hydrolysates—Food and Feed Additive and Use in Other Industrial Products

All compounds showing biological activity can be used as natural additives in modeling food products’ functional properties (Figure 5). Many groceries are susceptible to oxidative changes within unsaturated fatty acids [103]. These changes may harm product quality; sensory value can become significantly decreased with small oxidation changes. Additionally, lipid oxidation products (free radicals) may become toxic or carcinogenic [104]. Enzymatic protein hydrolysis is used to refine foods’ raw materials. In addition to the role that enzymatic protein hydrolysis fulfills in high-fat products, it is applied in food preparation technology to improve product consistency. The ingredients of hydrolysates have better emulsifying, foaming, and dispersing properties and solubility than parental proteins. In addition, hydrolysates also improve products’ taste and water absorption capacity [105]. Shortening protein chains usually improves nutritional value and enhances taste qualities. However, intense protein degradation can cause a bitter taste [106]. Mixtures of known, established, advanced, and designed hydrolysate compositions can be obtained with peptides’ size, quality, and share of free amino acids [107]. Additionally, using protein hydrolysates has economic justification, as treated products have a relatively extended shelf life. This method is often used in meat processing to increase the availability and attractiveness of the products for consumers [108]. However, protein hydrolysates have significantly wider use and are used in the feed industry for specific functions. They have a beneficial impact on performance traits and animal welfare due to their impact on gastrointestinal flora and feed digestibility. Protein hydrolysates are also used in the pharmaceutical and cosmetics industry (particularly in products with increased fat content) and the paper industry [109,110]. Protein hydrolysates are also increasingly used to produce paint, biodegradable materials, adhesives, binders, coatings, unique mechanical properties and barriers, nanomaterials, and biopolymers [111].

## 4. Unconventional and Alternative Sources of Proteins

Proteins obtained from alternative sources (Figure 6) such as plants and insects have attracted considerable interest in the formulation of new food products with a lower environmental footprint and offer a solution to feeding a growing world population. Unfortunately, there is little information available on these emerging protein sources, in contrast to the substantial amount of knowledge accumulated over the years regarding many established proteins and protein fractions.

### 4.1. Rice

Rice has the second-largest harvest and plant consumption in the world after wheat. Annual harvests oscillate around 480 million metric tons [112]. The highest consumption of rice is characteristic of Asian countries. Rice protein contains the second-highest lysine content (the limiting amino acid in cereal) [113] and is favored over other cereal proteins because of its amino acid structure. The protein content in rice is relatively low, at around 8%. However, it is one of the primary protein sources in southern countries and Southeast Asia due to high consumption [114]. In Europe and the United States, high-protein rice ingredients are increasingly used to produce gluten-free food due to their hypoallergenic properties [115]. The solubility of rice proteins is lowest at pH 4–5 and increases as the pH increases or decreases. Their high glutelin content mainly influences the ability of rice proteins to dissolve at certain pH values. Glutelin is the dominant protein fraction in endosperm and constitutes a significant proportion of all rice bran proteins [116]. Preparations based on rice proteins and their specific fractions also have antioxidant, antihypertensive, antineoplastic, and anti-obesity properties [117]. Rice proteins are becoming more and more popular in sports nutrition products and dietary supplements. This protein can be used as an alternative to the currently widely used casein, whey, or soy, and as an additive in the production of bread, biscuit, high-protein bars, or edible films, improving these products’ nutritional and functional value [117,118].

### 4.2. Corn

Corn is one of the most significant plants used in industry, particularly in the United States. The protein content in maize varies between 9% and 12%. Approximately 50% of the United States’ harvest is used as animal feed. The remaining percentage is used in spirits, syrup (glucose syrup production), maize flour, starch, oil, corn protein, and for many other applications. Additionally, in the food industry, maize is use intermediately in producing foodstuffs such as chips and tortillas [119]. Corn gluten flour is formed as a wet, milled maize product used for corn protein production. Corn gluten is one of the few vegetable proteins produced on an industrial scale, and is mainly used to produce polymers and food bags [120].

Zein is not popularly used as a source of protein in cornmeal and other products for human consumption. It exhibits poor water solubility and is deficient in certain essential amino acids such as lysine and tryptophan, limiting its application in the food industry. Consequently, it is mainly utilized as animal feed. Zein contains a high proportion of non-polar and hydrophobic amino acid residues buried inside the protein structure, responsible for its poor aqueous solubility [121]. Zein is corn’s major storage protein and comprises ≈45–50% of corn protein. The isolate obtained from zein is not used directly for human consumption due to its negative nitrogen balance and poor solubility in water. The ability of zein and its resins to form hard, glossy, hydrophobic, and fat-proof coatings and their resistance to microbial growth has attracted commercial interest. Potential applications of zein include fiber, adhesives, coatings, ceramics, ink, cosmetics, textiles, chewing gum, and biodegradable plastics. Zein has high potential in the specialty food, pharmaceutical, and biodegradable plastic industries, but only if manufacturing costs can be decreased [122]. Modifying this protein’s amino acid composition to permit its utilization in the food industry would increase its market value and range of applications [121].

The structural property of corn peptides is responsible for improved solubility. This property is mainly reflected in high solubility across a wide pH range, the formation of homogenous solutions with no precipitation, and the flow phenomenon. In addition, corn peptides have good solubility even under extreme conditions, such as low pH, so they are widely used in the acidic beverage industry. Furthermore, corn peptide drinks have the advantages of low viscosity, high glutamate content, and a pleasant taste and can improve brain function, making corn peptide beverages popular [123].

### 4.3. Quinoa

Quinoa (*Chenopodium quinoa* Willd) is an annual plant species. The bran fraction content in quinoa seeds is higher than in cereals such as wheat or corn, providing high protein and fat content in this plant’s seeds [124]. The amino acid composition of quinoa proteins is well-balanced and has a higher content of exogenous amino acids than most cereals [125]. Quinoa contains polyphenols, phytosterols, and flavonoids and is a rich source of dietary fiber, minerals, and vitamins. Due to its functional properties, solubility, emulsifying, foaming, and gelling properties, quinoa has various applications in the food and other, industry [126]. Quinoa oil is high in omega-6 and vitamin E content. Quinoa starch is used in many innovative industrial applications due to its functional properties, including the stability of its structure during freezing and the modification of solution viscosities [127].

### 4.4. Beans

Beans are a rich source of bioactive peptides, polysaccharides, oligosaccharides, and polyphenols. As a legume, beans have a naturally high protein content (up to approximately 30%). Bean protein is becoming more and more popular due to bioactive peptides, making it possible to use this protein to produce anti-diabetic, hypotensive, anti-inflammatory, and metal-chelating medicines [128]. In addition, beans contain a huge amount of dietary fiber and can improve exposure to irritation of the intestinal mucosa by facilitating the expulsion of toxic compounds in the intestines (especially in the large intestine). The high fiber content of beans can also help to reduce constipation and hemorrhoids and support easier defecation [129]. Legumes are a rich source of protein and dietary fiber. However, a wide variety of antinutritional factors, such as raffinose family oligosaccharides (RFOs), neurotoxins, proteinaceous compounds, lectins, goitrogenic factors, amylase inhibitors, and phytic acid, are present in them [130]. These factors influence their bioavailability and nutrient absorption in humans and animals eating beans as food [131]. Breeding crop varieties with a reduced concentration of antinutritional factors, using enzymes to reduce their concentration, and local methods such as cooking, germination, and soaking are all possible methods to reduce the antinutritional factors of beans [132].

### 4.5. Lupine

According to scientific research, lupine contains slightly higher protein content than other legumes commonly consumed by humans and is practically free of starch (up to 2%). Lupine’s structure is typically dicotyledonous. Its thick skin is approximately 30%, by weight of seeds, much higher than most domesticated species of cereals and legumes. The shell consists of cellulose and hemicellulose, and lupine seed fibers consist of soluble and insoluble fractions in the ratio of 40:60% [133]. The crude protein content of lupine fluctuates from 28% to 42%, depending on the type, variety rounds, conditions of growth, and soil type. In fractional lupine, protein composition is dominated by albumin and globulins (38% and 35%, respectively), 4.3% glutelin, and 0.6% prolamine. This combination of proteins is easily digestible, giving lupine a significant advantage over its competitors (soy, peas, and beans) [134]. In addition, products containing lupine are more easily digested than products containing soy or peas in their composition, as the fractional protein comprises lower proteolytic enzyme content inhibitors than other legumes. In addition to the full protein composition, lupine is a good source of vitamins, containing fat-soluble oils and provitamins, including sterols, carotenoids, and tocopherol, and much lower inhibitor content and water-soluble vitamins including riboflavin, thiamine, pyridoxine, folic acid, and ascorbic acid [135]. Protein preparations from lupine are successfully used in sports food formulas, bakery and confectionery recipes, and meat or dairy production technology. Many lupine products have been developed in recent years, including lupine oil, lupine protein concentrates and isolates, low-fat lupine flour and malt, and lupine flour extrudate [136].

### 4.6. Sunflower

Sunflower is one of the most produced oilseed crops, alongside soybean, rapeseed, cottonseed, and peanut [137]. According to the FAO, the world production of sunflower seed in 2019 was 56.07 Mt [138]. In summary, the whole sunflower seed contains 10–27% protein. However, when producing sunflower meal, the percentage increases to 40% for mechanically extracted seeds and 50% when the oil is removed with an organic solvent. In dehulled seeds, the protein percentage can reach 53–66% [139]. Sunflower proteins are primarily located in protein bodies and protein storage vacuoles of embryo and endosperm cells. Approximately 87–99% of the nitrogen in sunflower seeds corresponds to intact proteins. The remaining 1–13% originates from peptides, amino acids, or other nitrogenous substances. Sunflower’s total carbohydrate content ranges from 4% to 18%, and sunflower seed carbohydrates are characterized by a low starch content (around 0.42%) [140]. Sunflower protein composition complies with the FAO recommendations [141], except for its Lys content, and sunflower contains less sulfur-containing amino acids than rape protein, especially Met and Cys. The content of acidic amino acids (20%) and basic amino acids (18%) is relatively balanced. Unfortunately, using this protein in food products is limited due to its dark color and characteristic aftertaste. However, the uninteresting color can be easily masked by covering the product with chocolate or glaze. Similarly, the strength of the aftertaste depends on the concentration of sunflower protein. Therefore, mixing with other, better-tasting proteins can potentially reduce the negative aftertaste without lowering the protein content [118].

### 4.7. Insects

Insects are part of the traditional diet of approximately two billion people worldwide [142]. In some regions, insects have been part of the human diet for centuries, specifically as an alternative protein source, making them a subject of great interest. Human consumption of insects is associated with communities located in many parts of Asia, Latin America, and Africa [143]. Insects (invertebrates) possess huge biodiversity, and their biomass represents 95% of the animal kingdom [144]. They can be consumed in different life stages—eggs, larvae, pupae, or adults—and have been used as human food from prehistoric times. The main orders of consumed insects are Coleoptera (31%), Lepidoptera (18%), Hymenoptera (14%), Orthoptera (13%), and Hemiptera (10%) [145]. In May 2021, the Committee on Plants, Animals, Food and Feed (Novel Food and Toxicological Safety section), composed of representatives from all EU countries, gave a positive opinion of a draft legal act authorizing the sale of yellow mealworm (*Tenebrio molitor*) as a novel food [146]. Insects’ high protein levels are the main component of their nutrient composition, and they also possess significant amounts of other important nutrients such as lipids, beneficial fatty acids, vitamins, and minerals [147]. Compared to plant and meat proteins, insect proteins have high levels of high-quality nutritional protein, high total protein levels, and an essential amino acid profile of 50–80%. In general, insect lipids contain high amounts of unsaturated fatty acids relative to saturated fatty acids [148].

Furthermore, many minerals are found in insects, such as iron, zinc, potassium, sodium, calcium, phosphorus, magnesium, manganese, and copper. In addition, they contain a great variety of lipophilic vitamins and riboflavin, pantothenic acid, biotin, and, sometimes, folic acid [149]. Therefore, insect proteins are a promising raw material for further research and industrial use. Unfortunately, at this time, many doubts exist about using insects as food. Specifically considering safety concerns, the common hazards related to insect consumption are microbiological, parasitological, and allergenic. Therefore, the production technology and safety of their use require further research and testing [150].

### 4.8. Algae

Algae are a varied group of species described as oxygen-producing, photosynthetic, unicellular, or multicellular organisms, excluding embryophyte terrestrial plants and lichens. Depending on the type and place of harvest, marine algae may have a protein content of 20% to 60% of dry matter [151,152]. Due to their pigmentation, macroalgae can be divided into three main groups: *Chlorophyta* (green algae), *Phaeophyta* (brown algae), and *Rhodophyta* (red algae) [153]. Thus, microalgae are a hugely diverse group. Several species are exploited for various biotechnological purposes, such as biofuel and animal feed.

Furthermore, *Arthrospira platensis* (Spirulina) and *Chlorella vulgaris* (Chlorella) are sold as functional foods due to their high vitamin and mineral content [154]. Algae are generally regarded as a rich protein source, their composition meets FAO requirements, and they are often compared with other protein sources, such as soybean or egg [155]. Limiting amino acids in most algae species are tryptophan and lysine, whereas aspartic acid and glutamic acid constitute a relatively large proportion of total amino acids in many seaweed species, largely contributing to the distinctive ”umami” taste associated with seaweed [156]. Today, microalgae are typically consumed as a dietary supplement in tablets, powder, and capsules. However, they are also incorporated into several functional foods, including pasta, bread, biscuits, drinks, sweets, high-protein bars, and beer. For example, AlgaVia^®^ is a company offering algae products, producing a protein- and lipid-rich algal powder from *Chlorella protothecoides* [118,154]. High doses of algal protein in food cause a characteristic aftertaste and significant hardening of the product, especially high-protein bars, so may not have a use in similar products [118]. However, this type of protein functions efficiently as a feed additive, especially for poultry. Supplementing poultry feed with microalgae as a protein source can improve health, productivity, and value, demonstrated by various species, including *Chlorella* sp., *Arthrospira* sp., *Porphyridium* sp., and *Haematococcus* sp. Chickens fed with supplemented Spirulina were reported to have increased viability, improved overall health, and reduced cholesterol, triglyceride, and fatty acid plasma concentrations [157].

**Figure 5 polymers-13-02506-f005:**
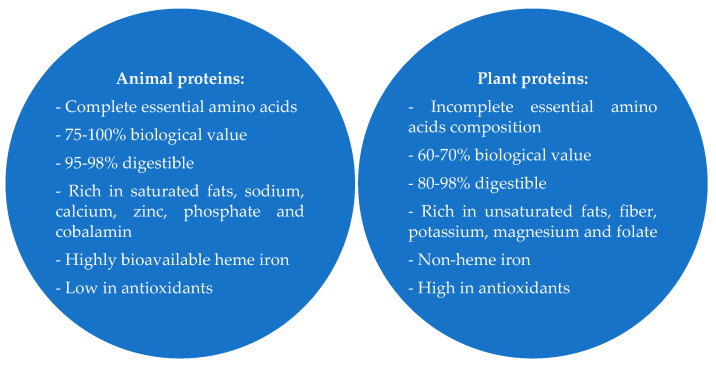
Animal and plant protein comparison based on Berrazaga et al., 2019 [158].

**Figure 6 polymers-13-02506-f006:**
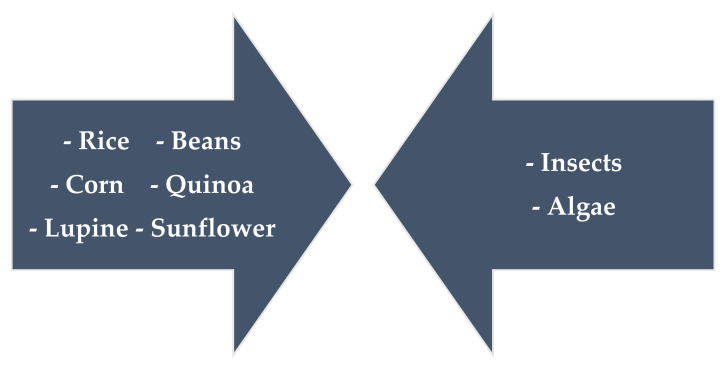
Unconventional and alternative protein sources are increasingly used in the food industry.

## 5. Conclusions

The human population is constantly increasing, and with it, the demand for protein. Therefore, we can expect an increasing demand for this nutrient in the coming years. In addition, the versatile, functional properties of proteins attract growing interest. Considering the increasing demand for proteins in the food industry and consumer trends for a healthy lifestyle, we can also expect an increase in interest in unconventional proteins such as lupine, bean, and other plant sources. However, an incomplete set of essential amino acids and issues with plants’ aftertaste are problems often associated with plant proteins. These aspects are obstacles in the use of these proteins as substitutes for conventional proteins.

Further development required to expand our knowledge of bionanomaterial production, including the food industry and biodegradable packaging materials, is challenging various fields of science. With the discovery of new nanoscale materials, new fields of application will emerge.

Insects are a promising alternative to conventional protein sources, having great potential as a component of the human diet due to their high nutritional value. However, the problems of lack of acceptance of insects as a foodstuff in developed countries and difficulties with introducing food products containing insects to the market remain. Additionally, using insects in the food industry on a large scale is challenging due to consumer safety issues, which must be confirmed by further research. Undoubtedly, using alternative sources of protein, especially edible insects, can solve the significant and growing issues of environmental and economic problems, and malnutrition. However, the globalization of insects and other unconventional protein sources in human nutrition undoubtedly requires efforts to increase public demand and acceptance and improve consumer awareness of the benefits of their consumption. Furthermore, the search for new insects and plants as sources of protein and the technology for their processing requires further research.

## Figures and Tables

**Figure 2 polymers-13-02506-f002:**
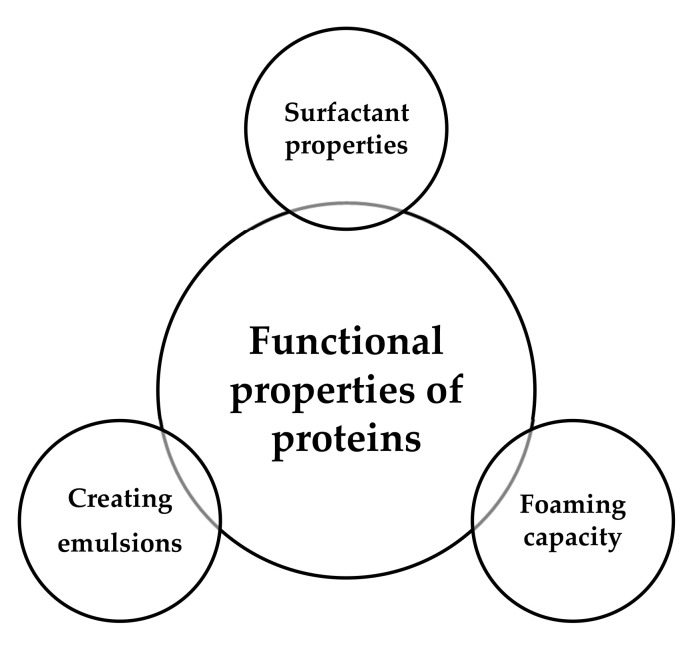
Classification of the main functional properties of proteins.

**Figure 3 polymers-13-02506-f003:**
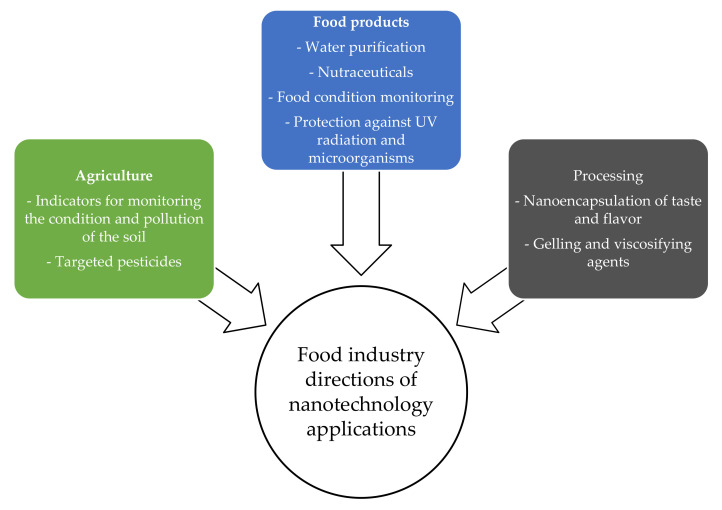
Directions of nanotechnology applications in the food industry based on [52].

**Figure 4 polymers-13-02506-f004:**
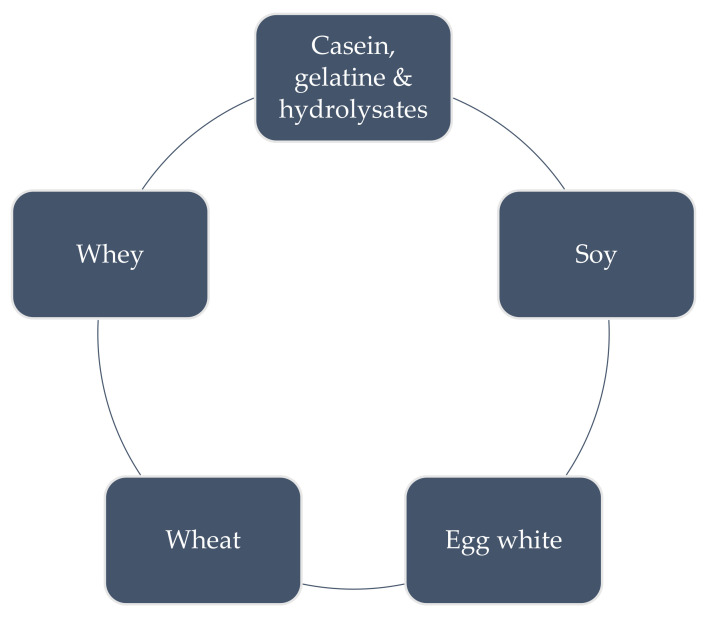
The most common sources of protein in food production.

## Data Availability

The data presented in this study are available upon request from the corresponding author.

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
