# Peer review of "Proteins in Food Systems—Bionanomaterials, Conventional and Unconventional Sources, Functional Properties, and Development Opportunities"

_polymers, 2021, doi:10.3390/polym13152506_

Round 1

Reviewer 1 Report

The work of Malecki describes the properties of conventional and unconventional proteins in food applications.

I found the work interesting but it should be carefully revised before its publication.

First, English language level should be seriously improved in the entire manuscript. The authors could ask a colleague who is proficient in English language to read their work. 

The major goal of this work is to describe utility and application of proteins from several sources, including some innovative ones, in the frame of food systems. I have seen that proteins are within the scope of the journal as biopolymers. However, in my opinion, Polymers' Readership would be more interested in reading applications of proteins in more fields. I strongly suggest to add one or more sections on protein applications in materials and in (bio)nanomaterials fabrication (theme treated in DOI: 10.1016/j.sbi.2020.04.005) The authors should thus, modify the title, abstract and conclusions of their work (including protein-based materials) that at the present seems more suitable for a more nutrition-oriented journal. Also protein and peptide assembly should be mentioned citing at least the works DOI:  10.1186/s12915-020-0751-4 and 10.3390/molecules26123558

line 33: proteins are not composed of dipeptides/oligopeptides: please better clarify and rephrase the sentence 'proteins are polypeptides containing two (dipeptides), four to ten (oligopeptides), and 33 more amino acids'

lines 35-37: unclear sentence: please rephrase 'It is caused by the multithreaded conditioned protein metabolism, whose specific feature is so-called: protein turnover,'

line 55 retinol (vitamin A): one would have expected to read in analogy to the previous (hemoglobin), (transferrin), (Vitamin A transport proteins) or something similar

line 60: it clearly refers to enzymes. In this case, mention enzymes in a clear form

line 72: amino acid distribution should be amino acid composition?

lines 108-109: unclear sentence, please rewrite: Using proteins increases everywhere where their high solubility is maintained.

lines 134-135: unclear, rewrite: The gel matrix is retained and immobilized water molecules and other food ingredients

lines 146-148: the sentence after ref. 31 seems not connected to the previous on or it needs a different link. Check and rewrite: 'as factors influencing the formation of foams can be mentioned: surface hydrophobicity, location of hydrophobic residues of amino acids on the surface of the protein, thiol groups, cations and anions, carbohydrates, lipids.'

section 2.4: provide examples of emulsions-creating proteins in this section

line 164: rewrite: 'It is a state the most energy-efficient'

line 248: correct 'processin point of view'

figure 3: the arrows suggest that each source in the figure is linked in someway to the others, but this is not reported in the text. Remove arrows.

section 3.5: report the protein percent in egg white (about 10%)

line 288: transporter, antibacterial or anti-cancer agent, provide references

lines 289-290: 'Ovalbumin is widely used as a dietary supplement and ovomucin as a cancer inhibitory agent.' please provide references

line 295: 'possible applications.' provide references

lines 298-300: unclear sentence, rewrite: 'in confectionery, confectionery in foam products, in the production of beer, wine, mead, as well as in the fat industry (oils)'

lines 303-304: unclear sentence, rewrite: 'Especially large the collagen content is characterized by connective tissue, bones, and'

line 329: 'Also it allows' Specify what 'It' refers to.

lines 344-345: should it be 'biodegradable materials'?

Between 3 and 3.1, and 4 and 4.1 you should provide a text in which you explain what will be treaded in the following sections.

lines 360-361: 'have antioxidant, antihypertensive, antineoplastic, anti-obesity properties.' provide references

line 374: introduce zein protein giving more details on it.

line 395: rewrite 'such compounds as'

lines 413-414: I read that lupine would have twice the protein contents of other legumes but in the previous section beans are said to have up to 30%, whilst lupines reach 42% (not 60%). Please check, and correct/rewrite.

line 425: 'much lower content inhibitors' a 'of' is missing.

Figure 4: the arrows are misleading (it seems that usage of some protein sources is decreasing with respect to algae and insects). If this is not clearly stated in the manuscript, arrows should be removed.

lines 450-452: rewrite. In fact, I do not understand the link of soybean and other specific amino acid-sources with sunflower. Clarify and rephrase.

section 4.8: provide typical protein percentage in Algae

Figure 5: arrows are misleading in my opinion. Remove them.

Line 515: rewrite 'the population of people' and 'proteins make they will'

Rewrite conclusions including importance also of protein-based materials and nanomaterials.

Author Response

University of Life Sciences in Lublin

Faculty of Food Sciences and Biotechnology

Department of Dairy Technology and Functional Foods

Skromna 8, 20-704 Lublin

Poland

Phone: +48 81 4623350

Fax:     +48 81 4623345

July 25, 2021,

Dear Reviewer,

Our manuscript entitled “Proteins in food systems – bionanomaterials, conventional and unconventional sources, functional properties, and development opportunities” is being re-submitted for publication in Polymers (Ref. No.: polymers-1301888 – Round 1) has been revised and is being re-submitted for publication in Special Issue Characterization, Bioactivities and Biotechnological Applications of Proteins and Peptides (Section Biomacromolecules, Biobased and Biodegradable Polymers).

We have carefully considered each of the comments and made the appropriate revisions in the manuscript. An itemized list of our responses to each of the comments is included below.

Thank you for your kind attention.

Yours faithfully,

Bartosz Sołowiej

Reviewer 1

Comments and Suggestions for Authors

The work of Malecki describes the properties of conventional and unconventional proteins in food applications.

I found the work interesting but it should be carefully revised before its publication.

First, English language level should be seriously improved in the entire manuscript. The authors could ask a colleague who is proficient in English language to read their work. 

The major goal of this work is to describe utility and application of proteins from several sources, including some innovative ones, in the frame of food systems. I have seen that proteins are within the scope of the journal as biopolymers. However, in my opinion, Polymers' Readership would be more interested in reading applications of proteins in more fields. I strongly suggest to add one or more sections on protein applications in materials and in (bio)nanomaterials fabrication (theme treated in DOI: 10.1016/j.sbi.2020.04.005) The authors should thus, modify the title, abstract and conclusions of their work (including protein-based materials) that at the present seems more suitable for a more nutrition-oriented journal. Also protein and peptide assembly should be mentioned citing at least the works DOI:  10.1186/s12915-020-0751-4 and 10.3390/molecules26123558 –

Thank you for your comment. A native American English speaker has read and corrected the above-mentioned manuscript. Also,  we have rectified and corrected the manuscript with regard to the Reviewer’s suggestions. Lines: 2-4; 18-21; 41-49; 225-254; 653-655; 662-665

line 33: proteins are not composed of dipeptides/oligopeptides: please better clarify and rephrase the sentence 'proteins are polypeptides containing two (dipeptides), four to ten (oligopeptides), and 33 more amino acids'

Thank you for your comment. We have clarified and rephrased the sentence. Lines: 35-39

lines 35-37: unclear sentence: please rephrase 'It is caused by the multithreaded conditioned protein metabolism, whose specific feature is so-called: protein turnover,'

Thank you for your comment. We have reformulated and corrected the incomprehensible sentence. Lines: 41-43

line 55 retinol (vitamin A): one would have expected to read in analogy to the previous (hemoglobin), (transferrin), (Vitamin A transport proteins) or something similar

Thank you for your comment. We have enriched the text with data on examples of proteins responsible for retinol transmission. Lines: 83-85

line 60: it clearly refers to enzymes. In this case, mention enzymes in a clear form

Thank you for your comment. We have added an excerpt on enzymes in pure form as per Reviewer's guidelines.  Lines: 93-97

line 72: amino acid distribution should be amino acid composition?

Thank you for your comment. We have corrected the spelling mistake. Line: 106

lines 108-109: unclear sentence, please rewrite: Using proteins increases everywhere where their high solubility is maintained.

Thank you for your comment. We have reformulated and clarified the sentence. Lines: 142-143

lines 134-135: unclear, rewrite: The gel matrix is retained and immobilized water molecules and other food ingredients

Thank you for your comment. We have improved and corrected the sentence. Lines: 168-171

lines 146-148: the sentence after ref. 31 seems not connected to the previous on or it needs a different link. Check and rewrite: 'as factors influencing the formation of foams can be mentioned: surface hydrophobicity, location of hydrophobic residues of amino acids on the surface of the protein, thiol groups, cations and anions, carbohydrates, lipids.'

Thank you for your comment. We have checked and rectified it in accordance with the Reviewer's requirements.  Lines: 182-184

section 2.4: provide examples of emulsions-creating proteins in this section

Thank you for your comment. We have given examples of proteins used for emulsification. Lines: 210-212

line 164: rewrite: 'It is a state the most energy-efficient'

Thank you for your comment. We have added an explanation of this topic. Lines: 197-201

line 248: correct 'processin point of view'

Thank you for your comment. We have corrected the spelling mistake.  Line: 345

figure 3: the arrows suggest that each source in the figure is linked in someway to the others, but this is not reported in the text. Remove arrows.

Thank you for your comment. We have improved the chart as suggested by the Reviewer.  Lines: 377-379

section 3.5: report the protein percent in egg white (about 10%)

Thank you for your comment. We have added information about protein content. Lines: 394-395

line 288: transporter, antibacterial or anti-cancer agent, provide references

Thank you for your comment.  We have added references. Lines: 401-403

lines 289-290: 'Ovalbumin is widely used as a dietary supplement and ovomucin as a cancer inhibitory agent.' please provide references

Thank you for your comment.  We have added references. Lines: 403-404

line 295: 'possible applications.' provide references

Thank you for your comment.  We have added references. Lines: 407-409

lines 298-300: unclear sentence, rewrite: 'in confectionery, confectionery in foam products, in the production of beer, wine, mead, as well as in the fat industry (oils)'

Thank you for your comment.  We have reformulated and completed the fragment of the text as suggested by the Reviewer. Lines: 411-414

lines 303-304: unclear sentence, rewrite: 'Especially large the collagen content is characterized by connective tissue, bones, and'

Thank you for your comment. We have improved and corrected the sentence. Line: 418

line 329: 'Also it allows' Specify what 'It' refers to.

Thank you for your comment. We have clarified what the sentence is about. Lines: 443-444

lines 344-345: should it be 'biodegradable materials'?

Thank you for your comment. We have corrected the spelling mistake. Line: 456

Between 3 and 3.1, and 4 and 4.1 you should provide a text in which you explain what will be treaded in the following sections.

Thank you for your comment. Between 3 and 3.1 and 4 and 4.1 we have provided a text where we have explained what will be covered in the following sections. Lines: 256-259; 459-464

lines 360-361: 'have antioxidant, antihypertensive, antineoplastic, anti-obesity properties.' provide references

Thank you for your comment.  We have added references. Lines: 478-479

line 374: introduce zein protein giving more details on it.

Thank you for your comment.  We have developed the issue as requested by the Reviewer. Lines: 499-507

line 395: rewrite 'such compounds as'

Thank you for your comment. We have corrected and clarified the sentence. Lines: 522-523

lines 413-414: I read that lupine would have twice the protein contents of other legumes but in the previous section beans are said to have up to 30%, whilst lupines reach 42% (not 60%). Please check, and correct/rewrite.

Thank you for your comment. We have reformulated and corrected the sentence. Lines: 548-549

line 425: 'much lower content inhibitors' a 'of' is missing.

Thank you for your comment. We have corrected the sentence. Line: 562-563

Figure 4: the arrows are misleading (it seems that usage of some protein sources is decreasing with respect to algae and insects). If this is not clearly stated in the manuscript, arrows should be removed.

Thank you for your comment. We have removed the arrows and modified the chart. Lines: 568-571

lines 450-452: rewrite. In fact, I do not understand the link of soybean and other specific amino acid-sources with sunflower. Clarify and rephrase.

Thank you for your comment. We have reformulated and corrected the sentence. Lines: 584-586

section 4.8: provide typical protein percentage in Algae

Thank you for your comment. We have included information about the percentage of protein in algae. Lines: 624-625

Figure 5: arrows are misleading in my opinion. Remove them.

Thank you for your comment. We have removed the arrows and modified the chart. Lines: 649-651

Line 515: rewrite 'the population of people' and 'proteins make they will'

Thank you for your comment. We have reformulated and corrected the sentence. Lines: 653-655

Rewrite conclusions including importance also of protein-based materials and nanomaterials.

Thank you for your comment. We developed the conclusions according to the Reviewer's suggestions.  Lines: 653-655; 662-665

Reviewer 2 Report

This review article presents information about role of proteins in the human body, characteristic of conventional and unconventional or alternative sources of proteins, nutritional value, functinal properties, and food industry use. The article is based on a review of world literature. References well chosen. Before publishing in Polymers, the paper requires additions and corrections. The list of proposed changes is given below:

General comments:

L8-11 add the initials of the name and surname of each co-author of the article (the same as in the Author Contributions chapter) and their e-mail

Detailed comments:

L49 + add information about the forecasted increase in the production of meat from farm animals, legumes and other protein sources; consumers looking for new products of a healthy and safe food with pro-health values

L192 add information on the current share of GMO soybean production and the threat to its use in the feeding of farm animals, the food industry, and human diets.

L211 add information about the global production of wheat grain, the largest producers, use for feed and food, amino acids limiting the nutritional value of wheat protein, undesirable effects of wheat gluten in the human diet. Contraindications for consuming wheat gluten.

L252+ Write something about casein intolerance to cows' milk. What proportion of the population is affected.

L280+ Write something about the chemical composition of the liquid egg content, the nutritional value of the eggs; importance in the human diet as a cheap source of animal protein; covering the demand for essential amino acids, vitamins and minerale; against the indications of their consumption.

L349 Write something about AA content of essential limiting amino acids

L403+ write something about contraindications, negative effects

L440 provide data for 2019 according to FAOSTAT

L464 Add information about the population size for which insects are food products. Approval of mealworm larvae in the EU as a food product

 L544+ add sections Institutional Review Board Statement; Informed Consent Statement

L609, 614 Please check the correctness of the surname and first names of the authors

L652 "Nauki InĹĽ. Technol." instead of current form

Author Response

University of Life Sciences in Lublin

Faculty of Food Sciences and Biotechnology

Department of Dairy Technology and Functional Foods

Skromna 8, 20-704 Lublin

Poland

Phone: +48 81 4623350

Fax:     +48 81 4623345

July 25, 2021,

Dear Reviewer,

Our manuscript entitled “Proteins in food systems – bionanomaterials, conventional and unconventional sources, functional properties, and development opportunities” is being re-submitted for publication in Polymers (Ref. No.: polymers-1301888 – Round 1) has been revised and is being re-submitted for publication in Special Issue Characterization, Bioactivities and Biotechnological Applications of Proteins and Peptides (Section

Biomacromolecules, Biobased and Biodegradable Polymers).

We have carefully considered each of the comments and made the appropriate revisions in the manuscript. An itemized list of our responses to each of the comments is included below.

Thank you for your kind attention.

Yours faithfully,

Bartosz Sołowiej

Reviewer 2

This review article presents information about role of proteins in the human body, characteristic of conventional and unconventional or alternative sources of proteins, nutritional value, functinal properties, and food industry use. The article is based on a review of world literature. References well chosen. Before publishing in Polymers, the paper requires additions and corrections. The list of proposed changes is given below:

General comments:

L8-11 add the initials of the name and surname of each co-author of the article (the same as in the Author Contributions chapter) and their e-mail

Thank you for your comment. We have added the initials of the name and surname of co-authors of the article and their e-mails. Lines: 7-14

Detailed comments:

L49 + add information about the forecasted increase in the production of meat from farm animals, legumes and other protein sources; consumers looking for new products of a healthy and safe food with pro-health values.

Thank you for your comment. We have added an explanation of this topic. Lines: 62-77

L192 add information on the current share of GMO soybean production and the threat to its use in the feeding of farm animals, the food industry, and human diets.

Thank you for your comment. We have added information on the current share of GMO soybean production and the threat to its use in the feeding of farm animals, the food industry, and human diets as suggested by the Reviewer. Lines: 262-275

L211 add information about the global production of wheat grain, the largest producers, use for feed and food, amino acids limiting the nutritional value of wheat protein, undesirable effects of wheat gluten in the human diet. Contraindications for consuming wheat gluten.

Thank you for your comment. We have added information suggested by the Reviewer in the text. Lines: 296-300; 302-305; 320-328

L252+ Write something about casein intolerance to cows' milk. What proportion of the population is affected.

Thank you for your comment. We have included information about intolerances. Lines: 350-355

L280+ Write something about the chemical composition of the liquid egg content, the nutritional value of the eggs; importance in the human diet as a cheap source of animal protein; covering the demand for essential amino acids, vitamins and minerals; against the indications of their consumption.

Thank you for your comment. We have checked and enriched it in accordance with the Reviewer's requirements. Lines: 381-394

L349 Write something about AA content of essential limiting amino acids

Thank you for your comment. We have included information about AA content of essential limiting amino acids. Lines: 468-470

L403+ write something about contraindications, negative effects

Thank you for your comment. We have enriched the text with the above information. Lines: 538-546

L440 provide data for 2019 according to FAOSTAT

Thank you for your comment. We have corrected and applied the FAOSTAT data from 2019. Lines: 574-575

L464 Add information about the population size for which insects are food products. Approval of mealworm larvae in the EU as a food product

Thank you for your comment. We have developed the issue as requested by the Reviewer.  Lines: 594-595; 602-606

L544+ add sections Institutional Review Board Statement; Informed Consent Statement

Thank you for your comment. We have added a section as per the Reviewer's instructions. Lines: 687-689

L609, 614 Please check the correctness of the surname and first names of the authors

Thank you for your comment. We have corrected the authors name and surname. Line: 776

L652 "Nauki InĹĽ. Technol." instead of current form

Thank you for your comment. We have corrected the information about the abbreviation of the journal name.  Lines: 849-850

Round 2

Reviewer 1 Report

The article quality has been improved